# Geographical, Temporal and Host-Species Distribution of Potentially Human-Pathogenic Group B *Streptococcus* in Aquaculture Species in Southeast Asia

**DOI:** 10.3390/pathogens12040525

**Published:** 2023-03-28

**Authors:** Wanna Sirimanapong, Nguyễn Ngọc Phước, Chiara Crestani, Swaine Chen, Ruth N. Zadoks

**Affiliations:** 1Veterinary Aquatic Animal Research Health Care Unit, Department of Clinical Sciences and Public Health, Faculty of Veterinary Science, Mahidol University, Nakhon Pathom 73170, Thailand; 2Faculty of Fisheries, University of Agriculture and Forestry, Hue University, Hue City 53000, Vietnam; 3Laboratory of Biodiversity and Epidemiology of Bacterial Pathogens, Institut Pasteur, 75015 Paris, France; 4Infectious Diseases Group, Genome Institute of Singapore, Singapore 138672, Singapore; 5Sydney School of Veterinary Science, Faculty of Science, University of Sydney, Camden, NSW 2570, Australia

**Keywords:** *Streptococcus agalactiae*, zoonosis, host-adaptation, ST283

## Abstract

Group B *Streptococcus* (GBS) is a major pathogen of humans and aquatic species. Fish have recently been recognized as the source of severe invasive foodborne GBS disease, caused by sequence type (ST) 283, in otherwise healthy adults in Southeast Asia. Thailand and Vietnam are among the major aquaculture producers in Southeast Asia, with GBS disease reported in fish as well as frogs in both countries. Still, the distribution of potentially human-pathogenic GBS in aquaculture species is poorly known. Using 35 GBS isolates from aquatic species in Thailand collected from 2007 to 2019 and 43 isolates from tilapia collected in Vietnam in 2018 and 2019, we have demonstrated that the temporal, geographical, and host-species distribution of GBS ST283 is broader than previously known, whereas the distribution of ST7 and the poikilothermic lineage of GBS are geographically restricted. The gene encoding the human GBS virulence factor C5a peptidase, *scp*B, was detected in aquatic ST283 from Thailand but not in ST283 from Vietnam or in ST7 from either country, mirroring current reports of GBS strains associated with human sepsis. The observed distribution of strains and virulence genes is likely to reflect a combination of spill-over, host adaptation through the gain and loss of mobile genetic elements, and current biosecurity practices. The plastic nature of the GBS genome and its importance as a human, aquatic, and potentially foodborne pathogen suggests that active surveillance of GBS presence and its evolution in aquaculture systems may be justified.

## 1. Introduction

In the 1960s, group B *Streptococcus* (GBS) or *Streptococcus agalactiae* emerged as a leading cause of neonatal infections in the USA and Europe [1]. According to a 2021 report, the burden of disease is now highest in sub-Saharan Africa and South Asia (WHO, 2021). In addition to neonatal sepsis, meningitis, and neurodevelopmental impairment, GBS contributes to pre-term births, stillbirth, and maternal sepsis [2]. Since the 1990s and 2000s, GBS has increasingly been recognized as a pathogen of non-pregnant adults, where it causes invasive disease, pneumonia, urinary tract infections, and skin and soft tissue infections [3,4,5]. The most recent addition to the suite of disease manifestations caused by GBS in humans is foodborne disease, which may lead to meningitis and osteoarthritis, as first recognized in Singapore in 2015 and attributed to sequence type (ST) 283 [6,7]. This ST had previously been linked to adult invasive disease in Hong Kong [5], where, as in Singapore, the emergence of adult GBS meningitis was noted in the late 1990s [8]. While GBS is an important pathogen, the organism is also carried asymptomatically by a considerable proportion of men and women, notably in the rectum, urethra/vagina, or pharynx (19%, 14%, and 4% of people, respectively [9].

Phylogenetic studies conducted in the wake of the outbreak of foodborne GBS ST283 disease in Singapore, which was traced to the consumption of raw freshwater fish [10,11], showed that the emergence of ST283 dated back to the 1980s when aquaculture was starting to intensify [6]. Around that time, GBS was first described explicitly as a pathogen of farmed aquatic animals, notably in cultured bullfrogs [12]. Outbreaks in finfish were not recognized until the 21st century, with some of the earliest reports relating to cultured pomfret and seabream in Kuwait [13,14]. Subsequently, outbreaks were described in farmed tilapia (*Oreochromis* spp.) in Colombia [15], Malaysia [16], and many other countries, including Thailand and Vietnam [17]. GBS has since become a major pathogen of tilapia, the third most commonly farmed fish globally based on tonnage [18]. The foodborne GBS ST283 outbreak in Singapore was not specifically linked to tilapia but the potential misidentification of fish species and cross-contamination post-harvest complicated the identification of the original source [6].

Although whole genome sequencing is increasingly used for the characterization of GBS in specialized research or public health laboratories, the most widely used typing systems for GBS are serotyping and multi-locus sequence typing (MLST), which is the system used to define ST [19]. Currently, ten serotypes are recognized, with four subtypes in serotype III, identified based on the capsular locus nucleotide sequence [20,21]. The number of STs continues to expand, with over 2000 STs recognized at the time of writing (https://pubmlst.org/bigsdb?db=pubmlst_sagalactiae_seqdef (accessed on 27 March 2023). In frogs and fishes, only three serotypes of GBS have been identified, whereby each serotype corresponds to a specific MLST-based group of closely related ST known as clonal complex (CC). Serotype Ia corresponds to ST7 and closely related ST, which together form CC7; serotype Ib corresponds to ST260 and closely related ST in CC552; and serotype III-4 corresponds to ST283 and closely related ST in CC283 [17,22,23]. Serotype Ib/CC552 has a global distribution, which is attributed to the international dissemination of tilapia broodstock in the 1970s and 1980s (Kawasaki et al., 2017), whereas serotype Ia/CC7 and serotype III-4/CC283 have more limited geographical distributions. They are predominantly found in China and Southeast Asia, with additional reports of CC7 from Kuwait [24] and CC283 from Brazil [25]. It has been speculated that the emergence of ST283 resulted from an initial spill-over from humans into fish after intentional or accidental exposure to human excreta, followed by the acquisition of a mobile genetic element that confers a survival advantage in fish and expansion in its new niche during the intensification of aquaculture from the 1980s onward (Delannoy et al., 2016; Barkham et al., 2019).

Compared to human GBS, serotype Ib/CC552 has a reduced genome size, and its occurrence has only been documented in frogs and fishes [17,26]. By contrast, CC7 and CC283 have been associated with disease in fish and in people [7,24]. ST7 has caused several cases of neonatal meningitis in Japan. It is also a multi-host pathogen of animals, including aquatic mammals (bottlenose dolphin) and several fish species (mullet, seabream, tilapia) [17,24]. Moreover, human ST7 from a clinical case of neonatal meningitis has been used to induce experimental infection in tilapia [27]. To date, there is no evidence for the natural transmission of ST7 from fish to humans or vice versa. By contrast, for ST283, foodborne fish-to-human transmission is well-documented in Singapore [10,11]. In addition, there is growing evidence that ST283 contributes to the burden of adult GBS sepsis in Thailand [6,28]. GBS infections in fish in Thailand, however, have been attributed predominantly attributed to ST7 [17,29], raising questions about the origin of human ST283 infections in the country. ST283 has been reported from fish in Vietnam [6], and trade in fish between the two countries offers a potential explanation [30].

The current study aimed to improve our understanding of the spatial, temporal, and host-species distribution of GBS in aquatic species in Thailand and Vietnam, which are among the four major producers of tilapia in East and Southeast Asia [31]. Without prospective surveillance systems, we did so through retrospective analysis of GBS isolate collections obtained by aquaculture experts during the investigation of disease outbreaks.

## 2. Methods

### 2.1. Samples and Isolates

In Thailand, GBS isolates (*n* = 35) were obtained from 2006 to 2019 (inclusive) from samples of aquatic animals submitted by farmers to the Veterinary Aquatic Animal Research and Healthcare Unit (VAARHU), Mahidol University, Thailand, or collected on-farm and transported to the VAARHU in transport media. They originated from 14 geographical locations, including 25 isolates from 9 provinces in Central Thailand, 8 isolates from 3 provinces in Eastern Thailand and 2 isolates from 2 provinces in Western Thailand. Host species included amphibians (East Asian bullfrog, also known as Chinese edible frog or Taiwanese frog; *Hoplobatrachus rugulosus*, *n* = 2) and finfishes, i.e., Nile tilapia (*Oreochromis niloticus* (Linnaeus, 1758), *n* = 13), red tilapia (*Oreochromis niloticus* × *Oreochromis mossambicus*, *n* = 16), Mekong giant catfish (*Pangasianodon gigas* (Chevey, 1931), *n* = 1) and giant sea perch (also known as Asian sea bass or barramundi; *Lates calcarifer* (Bloch, 1790), *n* = 3). Except for barramundi, which can be grown in fresh, brackish, or marine water, all were freshwater species. Samples of kidney, liver and brain tissue were plated on tryptic soy agar (TSA; BD Difco, Fisher Scientific, Loughborough, UK) and incubated aerobically at 28 to 30 °C for 24 to 48 hrs. Isolates from pure or dominant cultures on TSA were subcultured and initially characterized phenotypically. Gram-positive cocci that formed small (1–2 mm) white colonies on blood agar were non-motile, catalase-negative, serogroup B positive, beta-haemolytic, and CAMP-positive were archived at −80 °C as *S. agalactiae*. If the last two tests were negative or inconclusive, species identity was confirmed by polymerase chain reaction (PCR) as detailed under molecular identification.

In Vietnam, isolates were obtained from the brain or head kidney of clinically affected tilapia (*n* = 43) on nine fish farms across 4 provinces (An Giang, Dong Thap and Can Tho in the Mekong Delta and Thua Thien Hue along the Perfume River in central Vietnam) in 2018 and 2019. Clinical signs included exophthalmos (“popeye”), ascites and neurological symptoms (aberrant swimming). Samples were inoculated onto TSA and incubated aerobically at 28 to 30 °C for 24 to 48 hrs. Isolates were identified as *S. agalactiae* by colony morphology, Gram stain, and Lancefield testing using group B antibodies (Wellcogen Strep B Latex Agglutination Test, Thermo Fisher, Kent, UK).

### 2.2. Molecular Identification

Stored cultures were grown overnight at 37 °C on TSA to check for viability and purity. A single colony of each isolate was inoculated into trypticase soy broth and incubated overnight at 37 °C. Cultures were pelleted by centrifugation, and DNA was extracted from bacterial pellets using the QIAamp^®^ DNA Mini kit (QIAGEN, Valencia, CA, USA). Bacterial pellets were suspended in enzymatic lysis buffer (20 mM Tris-Cl pH 8; 2 mM Sodium EDTA; 1.2% Trition^®^X-100; 100 mg/mL lysozyme) and incubated at 37 °C overnight, and the remainder of the extraction was conducted as per the manufacturer’s instructions. Species identity of Thai isolates was confirmed by PCR, using forward primer gyrA_F (5′-GCACAATGGTGGTCATATCG-3′) and reverse primer gyrA-R (5′-ACGCGCTGGTAAAACAAGAG-3′), which target the DNA gyrase subunit A region [32]. The PCR mixture contained 2.5 µL of 10× PCR buffer, 0.5 µL of 10 mM dNTP, 0.5 µL of each forward and reverse primers, 1 µL of template DNA, 0.25 µL of Taq polymerase, and ddH_2_O to a final volume of 20 µL. The PCR program consisted of denaturation at 94 °C for 5 min followed by 35 cycles of 94 °C for 30 s, 55 °C for 30 s, 72 °C for 30 s, and finally 72 °C for 5 min before cooling down to 16 °C. Amplified products were analyzed by electrophoresis on 1% agarose gel, using staining with ethidium bromide for 5 min and de-staining with distilled water for 15 min before observation under UV light. A no-template negative control was included in the PCR and electrophoresis procedure.

### 2.3. Genetic and Genomic Analysis

Isolates were grown overnight in brain heart infusion broth and pelleted, followed by enzymatic lysis at 37 °C for 45 min. DNA was extracted from bacterial pellets using the DNeasy Blood and Tissue Kit (QIAGEN). DNA extracts were submitted to the Genome Institute of Singapore, the Agency for Science, Technology and Research, where whole-genome sequencing was conducted using established methods (Chau et al., 2017; Kalimuddin et al., 2017). The M220 Focused Ultrasonicator was used for genomic DNA shearing (Covaris, Woburn, MA, USA), followed by library preparation using the TruSeq Nextera XT DNA Library Preparation Kit (Illumina). Sequencing was conducted on a HiSeq 4000 sequencer (Illumina, San Diego, CA, USA), generating 2 × 151-bp reads. Raw fastq reads were used to check species identity using Kraken v.2.1.2 [33], and ST was called using SRST2 v.0.2.0 [34] and the *S. agalactiae* MLST database at https://pubmlst.org/organisms/streptococcus-agalactiae (accessed on 27 March 2023) [19,35]. In silico serotyping was conducted using GBS-SBG (GBS serotyping by genome sequencing) [20] and the method developed by Metcalf and colleagues [36], with confirmation using the method of Sheppard and colleagues when needed [37]. The presence of phage integrases was determined using the method of Crestani and colleagues [38], whereas presence of tetracycline resistance (TcR) genes and virulence genes *scp*B*-lmb* was determined using BLAST. Panaroo [39] and IQtree [40] were used to generate a maximum likelihood phylogeny for the isolates included in this study. The geographic origin of the isolates was mapped using ggplot in RStudio using R (v4.0).

## 3. Results

All isolates were confirmed to be *S. agalactiae* based on whole genome sequencing. Almost half of the Thai isolates belonged to ST283 (*n* = 16), with the remainder belonging to CC7 (*n* = 19, including 18 ST7 and 1 ST500). ST283 isolates belonged to serotype III according to the Metcalf method. GBS-SGB refined the serotype to III-1 or III-4 but without definitive allocation of any of the isolates to either serotype. CC7 isolates were identified by both methods as having serotype Ia, with some ambiguity for two isolates (Appendix A). One isolate was attributed to serotype Ia or serotype VII by both methods, and one isolate to serotype Ia or serotype III-4 by GBS-SBG. In Vietnam, most isolates (*n* = 38) belonged to ST283 and serotype III (Metcalf)/serotype III-1 or III-4 (GBS-SBG), with a minority of isolates characterized as CC552 and serotype Ib (both methods; *n* = 3) or CC7 and serotype Ia (both methods; *n* = 2).

Geographically, CC7 and ST283 were quite widely dispersed in Thailand, with detections in 12 and 9 locations, respectively (Figure 1). In seven locations with multiple isolates, the two clades co-existed. Temporally, too, CC7 and ST283 showed wide and overlapping distributions, with CC7 isolates detected from 2007 to 2019 (inclusive) and ST283 from 2006 to 2019 (inclusive). Both clades were isolated from amphibians (frogs), barramundi, Nile tilapia, and red tilapia, whereas the single isolate from a Mekong giant catfish belonged to ST283. In Vietnam, CC7 and ST283 were both found along the Mekong River in two or three provinces, respectively (Appendix A). Isolates from CC552 were detected on two farms along the Perfume River, whereas no CC7 or CC283 isolates were detected in that area. Because all isolates were collected from tilapia in 2018–2019, host species variation and temporal trends could not be analyzed for Vietnamese isolates.

Within CCs, isolates are clustered by country, albeit with some genetic heterogeneity within countries (Figure 2). Virulence genes *scp*B-*lmb* were exclusively detected in isolates from Thailand (*n* = 13) and only in genomes that contained phage integrases (Figure 2), i.e., GBSInt3 in ST7 or GBSInt1 plus GBSInt7 in ST283 (Appendix A). Conversely, tetracycline resistance determinant *tet*(M) was almost exclusively found in isolates from Vietnam, and often without detection of a phage integrase. Thirteen isolates contained phage integrases in the absence of *scp*B-*lmb*, with GBSInt1 plus GBSInt7 found in ST7 (*n* = 2; Thailand) as well as ST283 (*n* = 9; both countries) and GBSInt1 plus GBSInt6.1 detected in ST283 (*n* = 2; Thailand). Because isolates from Vietnam were clustered by farm, statistical analysis of associations between countries and genetic features of interest was not attempted.

## 4. Discussion

We demonstrate the wide geographic, temporal, and host-species distribution of GBS ST7 and ST283 in farmed aquatic species in Thailand and Vietnam. Both ST have previously been associated with disease in humans [6,24], and severe invasive disease due to ST283 has specifically been attributed to fish consumption in Singapore [7] and Laos [41]. Serotype III and, more specifically, ST283 is also associated with invasive disease and meningitis in Thailand, where the incidence of invasive GBS disease is increasing but where a specific link to fish consumption has yet to be demonstrated [28,42,43]. In our current study, ST283 was detected in GBS isolates from Thailand dating back to 2007, and found in multiple host species and provinces, implying that it is well-established among aquaculture species in the country. In Vietnam, ST283 was by far the most common type detected in fish farms along the Mekong River, the country’s main finfish aquaculture region. It has also been reported as a cause of human GBS sepsis in Vietnam, although in a lower proportion of cases than in Thailand, with 4 of 13 (31%) and 73 of 102 (72%) of invasive human GBS cases, respectively [6]. Interestingly, Vietnamese ST283 isolates lack *scp*B-*lmb*, which has been associated with the virulence of GBS in humans [44].

In many locations, both in Thailand and Vietnam, ST283 co-occurred with ST7, indicating that multiple introductions of GBS have occurred, implying poor biosecurity. When growing fish in river-based cage systems or land-based ponds filled with river water, prevention of the introduction of pathogens may be difficult, especially if pathogens can survive in water. Wastewater and pond water can be reservoirs of GBS [45,46] and exposure of aquatic animals via contaminated surface water has been described on multiple occasions [17,47]. Interestingly, ST283 was commonly found in river cage-based tilapia farms in Thailand but not in earthen pond systems [48]. There is no evidence yet of ST283 in cage-based tilapia farms along the Perfume River. Thus, there is an opportunity to protect the Perfume River from this strain of GBS through biosecurity measures. This could include the use of locally produced fry and fingerlings or sourcing of fingerlings from GBS or ST283 free areas or facilities, as is the current practice.

Along the Perfume River, STs belonging to CC552 were identified. CC552 is the most widely distributed strain of GBS in fish globally and has been found in farmed, wild, and hobby fish from Africa, Australia, South America, Europe and Asia [17,23,49]. Due to reductive evolution, loss of virulence genes, and inability to grow at mammalian body temperature, members of this CC are not human pathogens [17,26]. Although CC552 is widespread globally [23], it was not detected in Thailand. This is unlikely to be an artefact of the methodology because the primary culture was conducted at 28 °C and both haemolytic, CAMP-positive (CC7, CC283) and non-haemolytic, CAMP negative (CC552) isolates were eligible for inclusion in the study.

The most common type of GBS isolated from aquatic species in Thailand was ST7. This is consistent with some previous reports from the country [17,29]. In contrast, other reports describe the predominance of ST283 [46,48]. ST7 is also the predominant strain of GBS mentioned in English-language literature about aquaculture species from China [50,51,52]. This includes several fish species that were ST7-positive in Thailand (tilapia, giant seaperch), as well as other fishes, including but not limited to mullet (*Mugil cephalus*), silver perch (*Bidyanus bidyanus*), Japanese sea perch (*Lateolabrax Japonicus*), Jade perch (*Scortum barcoo*), striped bass (*Morone saxatilis*) and brindle grouper (*Epinephelus lanceolatus*) [51]. In the Philippines, serotype Ia, which can be considered to represent CC7 when isolated from fish, was the predominant type among farmed tilapia, with occasional detection of serotype Ib, which corresponds to CC552 [53]. By contrast, ST7 is rare in Vietnam. Drivers behind the observed distribution are unknown. Considering that most scientific literature to date is based on passive rather than active surveillance, i.e., dependent on reporting outbreaks of disease and mortality in fish, a complete picture of the spatial distribution and evolutionary origins of the different clades is yet to be established.

The widespread occurrence of GBS in aquaculture species may have implications for human health for several reasons, including food security, food safety, and antimicrobial resistance (AMR) [31]. With the growth of the human population, increasing urbanization and, until the COVID-19 pandemic, a rise in average disposable income in many Asian countries, the demand for animal protein as a highly nutritious component of human diets increases. Intensification of food production is needed to meet the growing demand, but it also leads to high stocking densities and an increased risk of infection. Additional pressure on the production system is exerted by sand and water extraction from rivers and deltas, needed for cement to build accommodation for the growing human population, and by sea level rises, salination and extreme weather events associated with climate change. This confluence of stressors may contribute to host-species jumping and the emergence of new pathogens. Evolutionary spill-over of GBS between host species has occurred more commonly from people to fish than from fish to people [54], and may be followed by spill-back from fish to humans via food, as shown for ST283 [7,10]. Additional events of this nature could exacerbate the situation, e.g., if ST7 from fish acquired virulence genes that made it more pathogenic to humans or if additional human GBS strains acquired mobile genetic elements associated with adaptation of GBS to fish [17,54]. Considering the carriage of GBS in the human gastro-intestinal and urogenital tract; the limited sanitary infrastructure and wastewater treatment capacity that exists in many countries; and the volume of culture, capture, handling and consumption of fish, there are multiple and frequent routes for exposure and transmission in both directions. The widespread and often indiscriminate use of antimicrobials to control disease in aquaculture could contribute to the selection for AMR in aquatic and human pathogens and commensals, with hotspots for antimicrobial use identified in Southeast Asia [55,56]. In our study, resistance genes for tetracycline were observed in Vietnamese isolates but not in Thai isolates. This is consistent with previous findings but without an obvious link to selection pressure from the antimicrobial use [6].

## 5. Conclusions

Our primary concern when conducting this study was the possibility of widespread prevalence of human pathogenic GBS in aquaculture species in Southeast Asia. Although we document the occurrence of ST7 and ST283 across a wider range of host species, years and locations than reported before, we did not detect the presence of a key human virulence factor, *scp*B-*lmb*, in any ST7 isolates nor in any Vietnamese ST283 isolates. This is mirrored, to our knowledge, by a lack of reports of fish-borne cases of human ST7 infection or, in Vietnam, a limited number of reports of human invasive ST283 infections, which contrasts with the situation for ST283 in Thailand. It is conceivable that the original spill-over from humans to fish was followed by the acquisition of MGE (locus 3) that allowed expansion in the aquatic population and loss of MGE (*scp*B-*lmb*) that provided a survival advantage to human but not aquatic GBS, increasing the fitness in the spill-over host and reducing the risk of spill-back to the original host. Considering the high prevalence of commensal GBS in humans, the impact of GBS on aquaculture species, and the high genome plasticity of GBS [54], new strains of GBS with hybrid virulence characteristics may emerge, and active surveillance of GBS strains in aquaculture systems is recommended.

## Figures and Tables

**Figure 1 pathogens-12-00525-f001:**
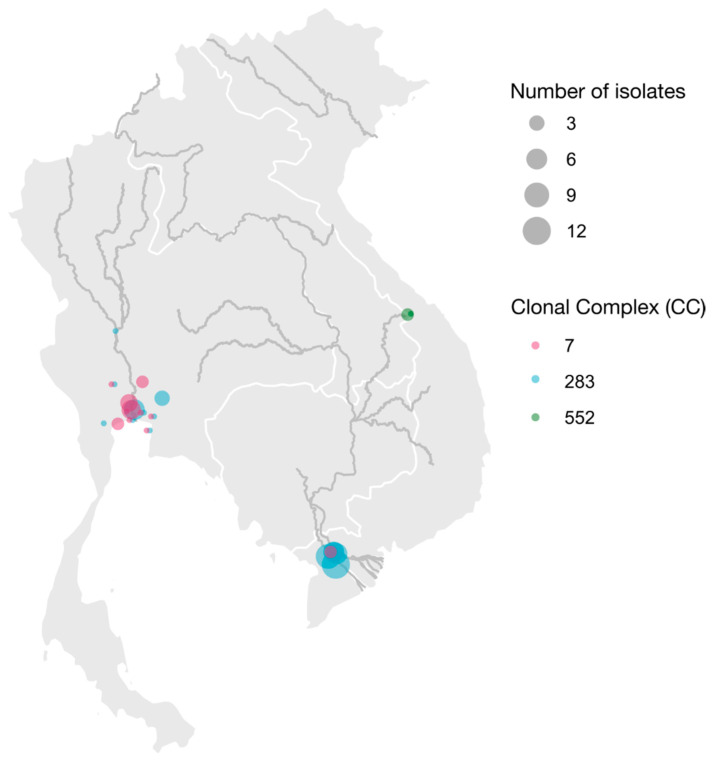
Geographic distribution of clonal complexes of Group B *Streptococcus* (*Streptococcus agalactiae*) isolates from farmed aquatic species in Thailand (**left**) and Vietnam (**right**). Dark lines indicate major waterways, white lines indicate country borders.

**Figure 2 pathogens-12-00525-f002:**
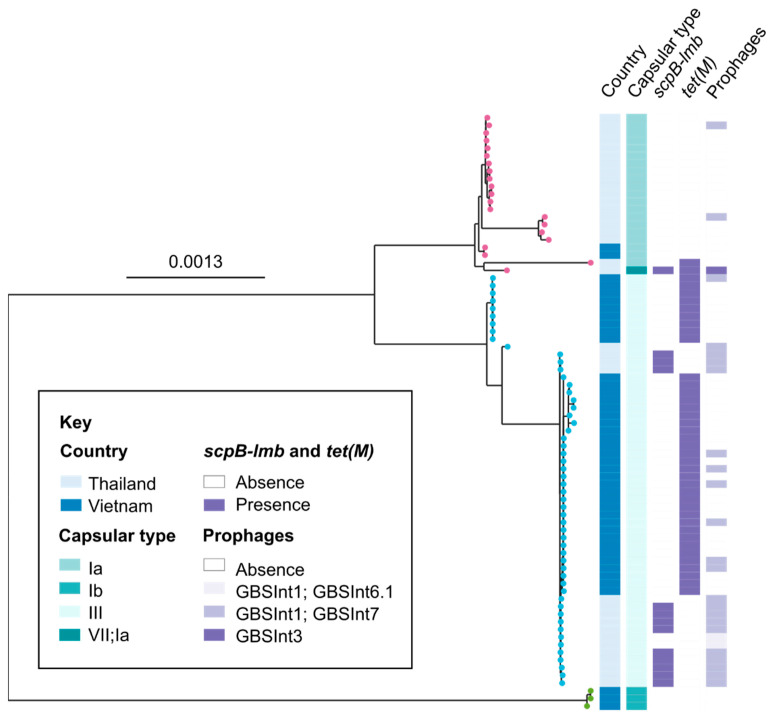
Maximum likelihood phylogenetic tree based on a core genome alignment of 78 Group B *Streptococcus* (*Streptococcus agalactiae*) isolates from aquatic species collected in Thailand (2007 to 2019) and Vietnam (2018 and 2019). Leaf colour indicates clonal complex (CC; pink = CC7; cyan = CC283; green = CC552). Strips indicate country of origin, capsular type derived from genomic sequence data, and other genetic features of interest. Tree was rooted at the midpoint.

## Data Availability

Sequence data are available in the European Nucleotide Archive Project PRJEB53664. Accession numbers for the ENA project and other supporting data are available in the Appendix A.

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
