# Peer review of "Geographical, Temporal and Host-Species Distribution of Potentially Human-Pathogenic Group B Streptococcus in Aquaculture Species in Southeast Asia"

_pathogens, 2023, doi:10.3390/pathogens12040525_

Round 1

Reviewer 1 Report

Regarding to manuscript 2247859, entitled” Geographical, temporal, and host-species distribution of potentially human-pathogenic Group B Streptococcus in aquaculture species in southeast Asia”, authors did genetic study of GBS fish isolates from two countries Tailand and Vietnam. However, authors did not include clinic GBS isolates from two countries for supporting their aims.

Several drawbacks of this manuscript shall be revised.

1.      There were only two frog isolates for 78 isolates. Authors may change the titles.

2.      Authors shall transfer the supplementary data into a Table.

-          Summary clinical information.

-          Add the human information of ST type in two countries.

-          Elucidate the genetic information related to host species, such as serotypes, antibiotic resistance related to integrin types.  

3.      Authors shall add the fish species as tilapia for those isolates from Vietnam.

4.      Authors shall check that Isolate number is 44 in abstract or 43 in methods.

5.      The isolation year differed between two countries and among fish species. Authors shall provide more evidence to support the “Title”.  

Author Response

Reviewer 1

Comments and Suggestions for Authors

Regarding to manuscript 2247859, entitled” Geographical, temporal, and host-species distribution of potentially human-pathogenic Group B Streptococcus in aquaculture species in southeast Asia”, authors did genetic study of GBS fish isolates from two countries Tailand and Vietnam. However, authors did not include clinic GBS isolates from two countries for supporting their aims.

AU: There are numerous studies supporting the role of GBS as human pathogen, including the role of GBS ST7 and GBS ST283, as detailed in the manuscript and supported by references. As explained in the introduction to the manuscript, GBS ST7 and GBS 283 are unique in that they have been found as a cause of disease in humans as well as fish. Other GBS strains have been found in humans but not in fish, or in fish but not in humans. The aim of this paper is to study the distribution of GBS in fish in Vietnam and Thailand and to determine, based on existing knowledge and typing methods, whether GBS from fish might pose a public health hazard. As acknowledged recently by the Food and Agriculture Organisation of the United Nations, diagnostic infrastructure for human and animal GBS is limited in Southeast Asia. This, in turn, limits our knowledge of the distribution of the potentially hazardous species or strains, which limits public and private incentives for investment in diagnostic infrastructure, which continues to limit knowledge. We are at risk of staying trapped in the vicious cycle of lack of awareness and lack of investment that plagues many neglected tropical diseases. With our contribution, we aim to raise awareness of this issue, and to expand our knowledge of the presence of potential public health hazards in aquaculture to stimulate further investment in diagnostics and research in this area. We hope that this will enable future studies along the lines suggested by the reviewer, where funding and infrastructure are made available for contemporaneously and sympatric prospective surveillance of GBS in aquaculture species and humans across Southeast Asia.

Several drawbacks of this manuscript shall be revised.

  1. There were only two frog isolates for 78 isolates. Authors may change the titles.

AU: Thank you for this suggestion. The title covers aquaculture species, which includes fishes and frogs, and does not state the number of fish, frogs, or animal species. As such, we consider the current title suitable.

  1. Authors shall transfer the supplementary data into a Table.

AU: The supplementary data is provided as a table in Excel rather than a fixed table in the text. The advantage of having the table as a supplementary Excel file is that readers can download the file and manipulate it, e.g., by selecting or sorting by host species, country, sequence type, tetracycline resistance, serotype, etc. We would like to provide readers with that option, which they would not have if we created a fixed format table in the text.

-          Summary clinical information.

AU: Thank you for that suggestion. This has been added in the Supplementary table.

-          Add the human information of ST type in two countries.

AU: We have included additional detail about ST7 in humans and animals in the introduction to explain the relevance of the study. We have provided more detail about ST283 in humans in Vietnam and Thailand in the discussion. We could not find information on ST7 in humans in Vietnam, Thailand, or elsewhere in Southeast Asia, which may reflect that this type is rare, or that there is a lack of routine genomic surveillance for GBS in the region, and no specific studies on ST7. We hope that our work demonstrating the widespread occurrence of ST7 in aquatic species may spark interest in ST7 and additional GBS surveillance in humans.

-          Elucidate the genetic information related to host species, such as serotypes, antibiotic resistance related to integrin types. 

AU: We have provided information about sequence types, serotypes, tetracycline resistance, and integrase typing in the supplementary table, which also includes information about host species. By including this as a supplementary file rather than a table, the reader can retrieve all relevant information for combinations that are of interest to them.

  1. Authors shall add the fish species as tilapia for those isolates from Vietnam.

AU: This information was covered in the text but not in the supplementary file. Thank you for pointing this out. We have now added this information.

  1. Authors shall check that Isolate number is 44 in abstract or 43 in methods.

AU: Thank you for that correction. We have updated the abstract to match the methods and results.

  1. The isolation year differed between two countries and among fish species. Authors shall provide more evidence to support the “Title”.

AU: There is growing concern about the emergence of zoonotic GBS in aquaculture species, whereby transmission to humans may happen via consumption of raw fish. This can lead to outbreaks, such the one observed in Singapore in 2015 and again, at a smaller scale, in subsequent years. It may also lead to an ongoing burden of sepsis in the country, as is likely to be the case in Thailand. The concern about the emergence of zoonotic GBS strains in aquatic species is such that the Food and Agriculture Organisation of the United Nations convened an expert panel to create a formal Risk Profile. In its summary, the FAO calls on readers to “Team up with the in-country experts” and states that there is “a pressing need to fill the data gaps identified.” – We have heeded this call by forming an international team of aquaculture experts, epidemiologists, and genomic scientists, to address the information gap around prevalence of zoonotic GBS in aquaculture in both countries. This is reflected in the title of the manuscript, and in the work we have conducted.

Reviewer 2 Report

The manuscript explores genetic characteristics of GBS isolates from aquatic species in Thailand and Vietnam. Also, geographical and host-species distribution is reported. Thailand and Vietnam are among the major aquaculture producers in Southeast Asia, with GBS disease reported in fishes as well as frogs in both countries.

Although the sample time period and the species identification was not the same in both groups, the results are interesting and the quality of the presentation is excellent.

Only minor details are here pointed out.  

Methods

Ln 116. Why for some specific names author names are added and for other ones, such as  Hoplobatrachus rugulosus, not?

Ln 138 and others. Stored cultures instead of archived cultures?

Molecular identification is only made for Thai isolates (Ln 144-156). Therefore, this section could be added below line 128. 

Ln 154. “Thai isolates”. Add Thai meaning the first time.

Author Response

Although the sample time period and the species identification was not the same in both groups, the results are interesting and the quality of the presentation is excellent.

AU: Thank you for your kind words. It is difficult to obtain funding for studies like this, and this has limited our ability to conduct a larger scale study across more years, host species or countries. We are glad that you see the value of our contribution, despite its limitations, and grateful for your positive feedback on the quality of the presentation.

Only minor details are here pointed out. 

Methods

Ln 116. Why for some specific names author names are added and for other ones, such as  Hoplobatrachus rugulosus, not?

AU: Thank you for that question. The names shown (Linneaous, Chiney, Bloch) are not the names of authors as we know them, but the names of the people that assigned species names to fish. In work on terrestrial animals, we don’t include those names routinely but it is commonly done in aquaculture. For some aquatic species, however, it is not known who named them (e.g., for the frogs) or their name is not a formal scientific name but the indication of a hybrid (e.g., for tilapia spp.)

Ln 138 and others. Stored cultures instead of archived cultures?

AU: We would use the word “archiving” for long term storage of cultures for future use and either word would seem to be correct here. We are happy to change this to “stored cultures” and have done so in the text.

Molecular identification is only made for Thai isolates (Ln 144-156). Therefore, this section could be added below line 128. 

AU: Thank you for that suggestion. In our original version, we had included this section under the information for Thai isolates. We did not like the flow of the text, however, as it jumped from isolates (Thailand) to molecular work (PCR) back to isolates (Vietnam) and then back to molecular work (sequencing). Therefore, we would prefer to maintain the current order of the text if you don’t mind.

Ln 154. “Thai isolates”. Add Thai meaning the first time.

AU: We are sorry, but we could not find any reference to Thai isolates in line 154.